# Factors associated with persistently high-cost health care utilization for musculoskeletal pain

Trevor A. Lentz[1]*, Jeffrey S. Harman[2], Nicole M. Marlow[3], Jason M. Beneciuk[4], Roger B. Fillingim[5], Steven Z. George[1]

1 Duke Clinical Research Institute and Department of Orthopaedic Surgery, Duke University, Durham, North Carolina, United States of America, 2 Department of Behavioral Sciences and Social Medicine, Florida State University, Tallahassee, Florida, United States of America, 3 Department of Health Services Research, Management, and Policy, University of Florida, Gainesville, Florida, United States of America, 4 Brooks Rehabilitation – College of Public Health & Health Professions Research Collaboration, Department of Physical Therapy, University of Florida, Gainesville, Florida, United States of America, 5 Pain Research & Intervention Center of Excellence, University of Florida, Gainesville, Florida, United States of America

* trevor.lentz@duke.edu

**Data Availability Statement:** The data underlying the results presented in the study are available from the Agency for Health Care Research and Quality (https://meps.ahrq.gov).

## Abstract

### Background

Musculoskeletal pain conditions incur high costs and produce significant personal and public health consequences, including disability and opioid-related mortality. Persistence of high-cost health care utilization for musculoskeletal pain may help identify system inefficiencies that could limit value of care. The objective of this study was to identify factors associated with persistent high-cost utilization among individuals seeking health care for musculoskeletal pain.

### Methods

This was a retrospective cohort study of Medical Expenditure Panel Survey data (2008–2013) that included a non-institutionalized, population-based sample of individuals seeking health care for a musculoskeletal pain condition (n = 12,985). Expenditures associated with musculoskeletal pain conditions over two consecutive years were analyzed from prescribed medicine, office-based medical provider visits, outpatient department visits, emergency room visits, inpatient hospital stays, and home health visits. Persistent high-cost utilization was defined as being in the top 15th percentile for annual musculoskeletal pain-related expenditures over 2 consecutive years. We used multinomial regression to determine which modifiable and non-modifiable sociodemographic, health, and pain-related variables were associated with persistent high-cost utilization.

### Results

Approximately 35% of direct costs for musculoskeletal pain were concentrated among the 4% defined as persistent high-cost utilizers. Non-modifiable variables associated with expenditure group classification included age, race, poverty level, geographic region,

**Funding:** This work was supported by the Foundation for Physical Therapy (https://foundation4pt.org) Promotion of Doctoral Studies I & II Awards to TAL; and the National Institutes of Health (NIH) Rehabilitation Research Career Development Program (https://www.nichd.nih.gov) K12-HD055929 to JMB. The funders had no role in study design, data collection and analysis, decision to publish, or preparation of the manuscript.

**Competing interests:** The authors have declared that no competing interests exist.

insurance status, diagnosis type and total number of musculoskeletal pain diagnoses. Modifiable variables associated with increased risk of high expenditure classification were higher number of missed work days, greater pain interference, and higher use of prescription medication for pain, while higher self-reported physical and mental health were associated with lower risk of high expenditure classification.

## Conclusions

Health care delivery models that prospectively identify these potentially modifiable factors may improve the costs and value of care for individuals with musculoskeletal pain prone to risk for high-cost care episodes.

## Introduction

Health care costs in the United States (U.S.) are concentrated among a small population with high levels of health care utilization.[1] These individuals are a target of health services research and policy because of their ability to expose potential system inefficiencies linked to low-value care.[2] Individuals with chronic musculoskeletal pain conditions may be particularly susceptible to high-cost utilization. High variability in care and a poor understanding of which pain treatments are most effective often lead to unnecessary care escalation, poor clinical outcomes, persistent health care needs, and avoidable opioid use.[3] The result is substantial cost associated with treatment of musculoskeletal pain (up to $650 billion annually).[4] While high costs are not always synonymous with low-value care, early identification and selective targeting of those at risk for persistently high costs due to musculoskeletal pain is an increasingly important priority of value-based systems that want to optimize distribution of health care resources.

Prior research has examined predictors of high-cost utilization; however, most studies assess high-cost utilization at one point in time. Identifying factors associated with persistence of high-cost utilization over time may be more relevant for those with musculoskeletal pain conditions because they are often recurrent, associated with increased medical comorbidities and result in chronic disability.[5] Another limitation of the existing literature is that most prior studies explore high-cost utilization in countries with government-funded, single-payer systems.[6–8] As a result, these studies may not translate well to health care systems like in the U.S. where enabling factors such as insurance status or socioeconomic status can have a profound effect on health care use and costs. To our knowledge, persistent high-cost utilization studies have not been performed in U.S. populations for highly-prevalent musculoskeletal pain conditions.

In this study, we identify sociodemographic, health, and pain-related factors that are associated with high-cost pain-related health care utilization over 2 consecutive years among individuals in the U.S. that receive health care for a musculoskeletal pain condition. Based on conceptual models of health care utilization[9–11] and existing literature on musculoskeletal pain outcomes, we hypothesize that high psychological distress, depression, pain interference, number of comorbid conditions, insurance type, and geographic region will be associated with persistent high-cost utilization.[9,12–15] Results will inform the development and testing of treatment pathways aimed at mitigating persistently high costs, with this analysis focusing on modifiable variables that could be targeted through treatment. Modifiable factors are those that can be reasonably changed during health care system encounters. In contrast non-modifiable factors are those that are not likely to change with a health system encounter. Examples of

modifiable factors would be prescription medication use while non-modifiable factors would be most sociodemographic characteristics and medical diagnoses. Identifying modifiable variables is a critical first step toward addressing recent national and global health initiatives calling for improved care of musculoskeletal pain conditions.[3,16,17]

## Materials and methods

### Dataset

This study used Public Use File Household Component data from the 2008–2013 (Panels 13–17) Medical Expenditure Panel Survey (MEPS), a set of large-scale surveys of families and individuals, their medical providers, and employers across the U.S. The survey includes data on demographics, health conditions, health status, use of medical care services, charges and payments, access to care, and health insurance coverage.[18] Subjects are enrolled in panels, each of which includes data collection in 5 rounds over 2 calendar years. MEPS uses a stratified, multistage sampling design that allows for nationally representative estimates of the U.S. noninstitutionalized civilian population.[19] In other words, MEPS strategically samples individuals so that by using the appropriate statistical procedures, the survey results can be used to make unbiased inferences about the target population. The University of Florida Institutional Review Board approved this study (IRB201600731) as exempt because it involved the use of existing, publicly available, de-identified data (https://meps.ahrq.gov/).

### Study sample

We identified survey respondents with musculoskeletal pain conditions using medical conditions files, which include diagnosis-level detail on medical conditions reported by MEPS respondents in each calendar year. To identify medical conditions, the survey asks respondents to report specific physical or mental health problems during the interview reference period, regardless of whether they sought medical care for these problems. Respondents are further required to identify conditions that are associated with health care events they report, or are the cause of missed school or work days.[18] Medical conditions are recorded as verbatim text and then coded to fully specified International Classification of Diseases, 9th Revision, Clinical Modification (ICD-9-CM) codes.

We considered respondents who reported at least one musculoskeletal condition in the index year of the panel (Year 1) for inclusion in the study. Selection of ICD-9 codes to be included in the study was informed by The Burden of Musculoskeletal Diseases in the United States: Prevalence, Societal and Economic Costs (BMUS), 3rd edition.[20] Our intent was to identify high expenditures for musculoskeletal conditions where expenditures are not expected to be persistently high in the majority of cases. Therefore, we excluded medically complex musculoskeletal conditions that typically produce high expenditures over prolonged periods, such as spinal cord injury, amputation, congenital deformities and cancer. In cases of uncertainty regarding inclusion or exclusion of a diagnosis, we generally erred on the side of inclusion to enhance generalizability of the results. The final list was decided upon by consensus of the authors. The ICD-9 codes included and excluded from the analytic sample are provided in S1 Table.

Respondents were excluded from the final analytic sample if they were not in scope for the entire panel, <18 years of age upon entry into the survey, did not provide data for all 5 rounds of the panel, were ineligible to complete the Adult Self-Administered Questionnaire (SAQ) in Year 1 or had a proxy complete the Adult SAQ. The Adult SAQ is a supplemental paper questionnaire administered to all household respondents 18 years old and older that includes questions from the Consumer Assessment of Health Plans (CAHPS®), and the SF-12. Our analysis

used multiple items from the SAQ. We excluded respondents with a proxy response to the SAQ because it measures subjective components of physical and mental functioning, depression and general psychological distress that are best completed by the individual respondent. [21–23] We also excluded those with incomplete data on the SAQ, which comprised approximately 3% of the sample. Finally, we also excluded respondents who reported zero expenditures in Year 1. Generally, those with zero expenditures reported a musculoskeletal condition in the survey, but did not receive care for the condition. Since we intended that results would inform clinical and health policy decision-making, limiting the analysis to a health care-seeking sample with 2 years of expenditure data was most appropriate.

## Expenditure summaries

The dataset includes separate, full-year event files for prescribed medicine, office-based medical provider visits, outpatient department visits, emergency room visits, inpatient hospital stays, and home health visits. Health care events were linked to each musculoskeletal condition reported by the respondent through the condition-event crosswalk, which is a numeric variable unique to each health condition that can be used to match conditions with their associated treatments throughout the survey. We then constructed annualized direct cost summaries for each condition that included payments from all sources for each event.[24] Finally, we developed person-level overall expenditure summaries for all musculoskeletal pain conditions for each of the 2 years in the panel. For Year 2 expenditure estimates, we only considered events linked to conditions reported in Year 1. For Year 2, summary costs were set at $0 if respondents reported no events for a condition. For each year, we assigned an expenditure percentile rank to each respondent based on overall expenditure summaries for the entire sample for that year.

## Model variables

We used the Value Model of Musculoskeletal Pain as a conceptual model to guide selection of sociodemographic factors, health-related factors, and pain condition characteristics to include in the analysis.[11] This conceptual model outlines processes by which health care system, provider and patient-level characteristics interact to drive health care costs and quality for individuals with musculoskeletal pain. The MEPS dataset includes extensive patient-level characteristics, but limited characteristics related to the health care system and provider for specific services or medical conditions. Therefore, model variables focused primarily on patient-level characteristics. Unless otherwise noted, all selected variables were from Year 1 of the longitudinal panel data file.

**General sociodemographic information.** Age, sex, race, ethnicity, years of education, body mass index (BMI), smoking status, poverty category, employment status, metropolitan statistical area (MSA), and census region.

**Health insurance coverage.** Medical Expenditure Panel Survey (MEPS) provides information on monthly payer status for each of the following: TRICARE, Medicare, Medicaid / State Children's Health Insurance Program (SCHIP), or private insurance. MEPS also includes summary measures that indicate whether a person has any insurance in a month. Each respondent was categorized as being privately insured all year, publicly insured all year, uninsured all year, or uninsured part of the year and either privately or publicly insured the remainder.[25]

**Usual care provider.** For usual source of care, the MEPS Household Component access-to-care section asks respondents whether there is a particular doctor's office, clinic, health center, or other place they usually go when they are sick or in need of health advice. We included this access-to-care measure because having a usual care provider has been associated with

reduced emergency room visits in studies of frequent health care utilizers and may have an important impact on costs.[26]

**Comorbidities.** We used the Deyo adaptation of the Charlson Comorbidity Index to determine comorbidity burden.[27] The index adaptation accounts for disease severity and comorbid conditions in studies of outcome and resource use employing administrative databases. The presence of 17 comorbid conditions was determined from International Classification of Diseases, 9th Revision (ICD-9) codes in respondents' medical condition files for Year 1. See Supplementary Material for a full list of conditions included in the index.

**General health status.** The SF-12 was used to assess general health status. The SF-12 is a self-reported health survey commonly used in musculoskeletal populations[28] that assesses eight domains of mental and physical health, including physical functioning, role limitations due to physical problems, bodily pain, general health perceptions, energy and vitality, social functioning, role limitations due to emotional problems, and mental health.[29] Mental and Physical Component sub-scores were calculated for analysis. Higher scores on this measure indicate better physical and mental health.

**Perceived health status and attitudes about health.** The survey asks respondents to report their perceived physical and mental health status compared to others. The survey also asks respondents if they felt they could overcome illness without help from a medically trained person. We used the latter response as an estimate of self-efficacy for managing one's own health. Self-efficacy is an important, potentially modifiable characteristic that can influence pain-related health care utilization and outcomes.[30] Although this question was not specific to self-efficacy for management of pain, we believed the question could provide insight into general confidence for managing health and illness, which could have an important impact on this intensity of health care utilization sought for their pain condition.

**General psychological distress.** The Adult Self-Administered Questionnaire (SAQ) includes six mental health-related questions, using the "K-6" scale developed by Kessler and colleagues.[31] Developed for use in the annual U.S. National Health Interview Survey and National Household Survey on Drug Abuse, the K-6 scale assesses non-specific psychological distress during the past 30 days and can be used to screen for individuals with mental illness at the population level.[32] Higher values indicate greater psychological distress.

**Depression.** The Adult SAQ includes two additional mental health questions from the Patient Health Questionnaire (PHQ-2).[33] These questions assess the frequency of the respondent's depressed mood and decreased interest in usual activities. The PHQ-2 is often used to assess depressive symptoms in patients with musculoskeletal pain.[34] Higher scores indicate higher levels of depressive symptoms.

**Days of work missed due to illness.** Sick leave can substantially contribute to the economic burden of musculoskeletal pain[4], and has been shown to predict future sick leave.[35] To measure sick leave in this study, we recorded the number of times the respondent lost a half-day or more from work because of illness, injury, or mental or emotional problems for each of the 5 rounds in Year 1 and summed for total days of work missed.

**Pain interference.** Pain interference with work and daily activities was assessed using the following SF-12 question from the Adult SAQ: "During the past 4 weeks, how much has pain interfered with normal work outside the home and housework?" This item is commonly used as a measure of pain interference in population-based studies for musculoskeletal pain conditions.[36] Higher values indicate higher pain interference.

**Diagnosis type.** We designed the study eligibility criteria to reduce the likelihood that medical complexity would substantially influence costs. However, certain diagnoses, such as fractures, sprains, and other acute injuries, are likely to be associated with higher initial costs, particularly if individuals undergo surgery. As a result, those with severe musculoskeletal

injuries might be more likely to be classified as high-cost utilizers, despite the appropriate and necessary use of high-cost interventions. To explore the effects of diagnosis type in the model, we identified two types of diagnoses based on ICD-9 code specifications: 1) Diseases of The Musculoskeletal System and Connective Tissue diagnosis only (ICD-9 codes 715–739), and 2) Musculoskeletal Injury (ICD-9 codes 805–959). For the purposes of analysis, we classified respondents into a "musculoskeletal disease only" group or a "musculoskeletal injury with or without a musculoskeletal disease" group.

**Total number of musculoskeletal pain diagnoses.** To control for the effect of multiple musculoskeletal conditions, we developed a summary count of musculoskeletal pain conditions reported in Year 1 of the panel. We identified these conditions by ICD-9 code using the same methodology as we did to identify eligible participants for the study.

**Number of prescription medications for pain.** The ongoing opioid crisis has prompted efforts to better understand how prescription pain medication use influences outcomes and costs across pain conditions.[37] Total number of prescription medications (opioid and non-opioid medications combined) linked to events for the index musculoskeletal pain condition were summed for Year 1 and included as a model variable. Medications could have been prescribed primarily for pain relief (e.g., NSAIDs or opioids) or may have been prescribed for other indications related to the musculoskeletal condition (e.g., muscle relaxants, psychotropic medication). Number of prescriptions is not the number of unique drugs per subject, but rather refers to the total number of prescriptions filled, which includes original fills and refills.

## Model variable classification

Variables were classified as modifiable or non-modifiable for the purposes of this analysis. We made this determination by considering the extent to which these variables could be reasonably modified through health care system encounters. We chose to classify variables in this way to inform future intervention studies. Interventions focusing on addressing modifiable variables could have the greatest potential to directly impact health care use and costs.

Non-modifiable variables are those that are less amenable to change through health care interventions or cannot be modified. Non-modifiable variables included most sociodemographic variables, such as age, sex, race, ethnicity, years of education, poverty category, employment status, metropolitan statistical area, census region, and insurance status. Non-modifiable general health- and pain-related variables included number of comorbidities, diagnosis type and total number of musculoskeletal pain diagnoses.

Modifiable variables are those that can be reasonably changed through health care system encounters. Modifiable variables included body mass index (BMI), smoking status, and presence of a usual health care provider. Self-reported general mental and physical health status, perceived health status compared to others, attitudes about health, levels of psychological distress and depression, and days of work missed due to illness were also included. Modifiable pain-related variables included level of pain interference and number of prescription medications for pain. Several variables such as days of work missed and number of prescription pain medications could be classified as either modifiable or non-modifiable, as they may often be viewed as an outcome of the condition. However, we categorized these as modifiable because they have been shown to change when included in studies as treatment targets.[38,39] For instance, moderate to strong evidence indicates that health-focused, service coordination, and work modification interventions that appropriately limit the days of work missed can significantly reduce work related disability due to musculoskeletal pain-related conditions and positively impact cost outcomes.[39] For this reason, we wanted readers to consider the modifiable potential of these variables for future analyses or when designing treatment programs.

## Statistical analysis

**Development of expenditure classifications.** We classified respondents into one of three groups depending on their expenditure levels in Year 1 and Year 2. Those in the top 15% of expenditures (85[th] percentile) specific to events with a musculoskeletal diagnosis in both years were defined as the "high health care expenditure (HIGH) group".[40–42] We defined those in the bottom 15% of expenditures (15[th] percentile) across both years as the "low health care expenditure (LOW) group". All others (remaining 70%) were defined as the "medium health care expenditure (MEDIUM) group". We allocated respondents who had expenditures in Year 1 of the panel and zero expenditures in Year 2 of the panel to the LOW or MEDIUM group based on their percentile expenditure from Year 1. For descriptive statistics, we used the medical care consumer price index to adjust all cost data to 2013 values.[43]

**Partially-adjusted multivariable models.** We took a two-step approach to identifying factors associated with persistent high cost utilization. First, we developed partially-adjusted multivariable models to examine the relationship of individual modifiable variables with expenditure group classification after controlling for non-modifiable variables. This approach would serve two purposes: 1) permitting us to understand the unique influence of individual variables with minimal covariate adjustment, and 2) informing the development of fully-adjusted multivariable models. We initially examined the appropriateness of both multinomial and ordinal regression given the nature of our outcome. Initial assessments indicated the data did not meet the parallel lines assumption, meaning that the single coefficient estimate provided by ordinal regression would not accurately reflect the true coefficients associated with moving from both LOW to MEDIUM and MEDIUM to HIGH membership. However, the assumption of the independence of irrelevant alternatives (IIA), which assess the appropriateness of multinomial regression was also not met, largely because we are evaluating an ordinal variable where LOW membership will be a closer substitute to MEDIUM membership than it will be to HIGH membership. In choosing between two imperfect models, we chose the model that produces a separate coefficient for each category, and is more interpretable in this context. Therefore, we used a multivariable generalized logit (multinomial) model analysis to determine factors associated with group classification for all multivariable models. For the partially-adjusted models, we developed a separate model for each modifiable variable adjusting for all non-modifiable variables.

**Fully-adjusted multivariable models.** Second, we developed fully-adjusted multivariable models including all non-modifiable variables and those modifiable variables that were significant in the partially-adjusted models. The fully-adjusted model also used a multivariable generalized logit (multinomial) model analysis to determine factors associated with group classification. To assess multicollinearity in the fully-adjusted model, variance inflation factor (VIF) and tolerance were calculated for each variable.[44] Missing values accounted for <5% of observations for all study variables. Therefore, rather than imputing missing variables, we used listwise deletion to analyze available data. The Wald chi-square statistic was examined as a measure of overall model fit. We compared adjusted risk ratios (RRs) and 95% confidence intervals (CIs) to determine the relative strength of each model variable. All models used the HIGH group as reference. However, we planned to report the inverse of the adjusted RR's so that higher risk of being in the HIGH group were associated with higher values of the model variables.

**Sensitivity analyses.** We selected a 15[th] percentile threshold to define persistent high cost utilization since this threshold has been commonly used in prior studies.[40–42] However, results could be sensitive to how persistent high cost utilization is defined, and therefore impact the interpretation of our findings. Therefore, we tested robustness of the final, fully-

adjusted model results using sensitivity analyses that alternatively defined group classification using 90th/10th and 80th/20th (HIGH/LOW) percentile thresholds.[6,42,45,46]

All regression and cost estimation analyses were conducted using survey procedures (PROC SURVEYLOGISITIC and PROC SURVEYMEANS; SAS v.9.3, SAS Institute Inc., Cary, NC) to account for the complex sampling design of the survey and employed person-level SAQ sampling weights to adjust for questionnaire non-response.[47] Taylor series linearization was used for variance estimation. Alpha was set at p = 0.05 for all analyses.

## Results

### Descriptive analysis

Of the 85,484 MEPS respondents considered, 13,332 had a musculoskeletal pain condition for which they had expenditures in the first year and self-completed the SAQ. Of those 13,332 respondents, 347 (2.6%) did not have complete follow-up data available for the entire panel and were excluded from the analysis. The final analytic sample included 12,985 respondents who met inclusion criteria over 5 panels. After applying respondent-level weights that account for the complex survey design and non-responders, those included in the final analytic sample represented approximately 150,792,698 (95% CI = 144,238,635–157,346,759) unique individuals in the U.S. noninstitutionalized civilian population during the study period. An outline of the selection process for the final analytic sample is provided in Fig 1.

Those with a proxy response on the SAQ (and therefore not included in the final analytic sample) differed from those self-completing the SAQ across numerous demographic and health-related variables. In general, those with a proxy response were more likely to be older, have more comorbid conditions, report higher pain, psychological distress, and depression, and lower physical and mental health than those completing the SAQ themselves. Specific variables that differed significantly between these groups are listed in S2 and S3 Tables, and implications of these differences for study interpretation are provided in the discussion.

The HIGH expenditure group consisted of n = 498 or 3.8% (3.5–4.3%), representing 5,896,762 (5,230,942–6,562,581) individuals in the U.S. noninstitutionalized civilian population after weighting. The MEDIUM group included n = 10,983 or 84.6% (84.7–86.3%), representing 128,931,988 (123,011,730–134,852,195) individuals after weighting. The LOW group included n = 1,504 or 11.6% (9.9–11.3%) representing 15,963,948 (14,815,815–17,112,081) individuals after weighting. Tables 1 and 2 report demographic and health-related information stratified by expenditure group, and for the entire sample.

We observed significant group differences for all variables except sex, BMI, smoking status, geographic region, education, and metropolitan statistical area. Diagnosis code frequencies for the sample are listed separately for diseases of the musculoskeletal system (S4 and S5 Tables). Adjusted mean (95% CI) and median (95% CI) annual musculoskeletal expenditures were $23.28 ($20.41–23.76) and $22.08 ($20.41–23.76) for the LOW group, $1,551.45 ($1,450.53–1,652.36) and $453.37 ($432.68–474.06) for the MEDIUM group, and $13,572.00 ($12,195.74–14,948.61) and $8,594.83 (7,842.09–9,347.57) for the HIGH group (Fig 2). Approximately 35% of the total weighted 2-year costs were attributable to the HIGH group, 65% were attributable to the MEDIUM group, and <0.1% were attributable to the LOW group. Distribution of musculoskeletal pain expenditures attributable to each event type per year for the three groups is presented in Fig 3.

### Partially-adjusted multivariable analysis

Non-modifiable variables associated with expenditure group classification included age, race, poverty level, geographic region, insurance status, diagnosis type and total number of

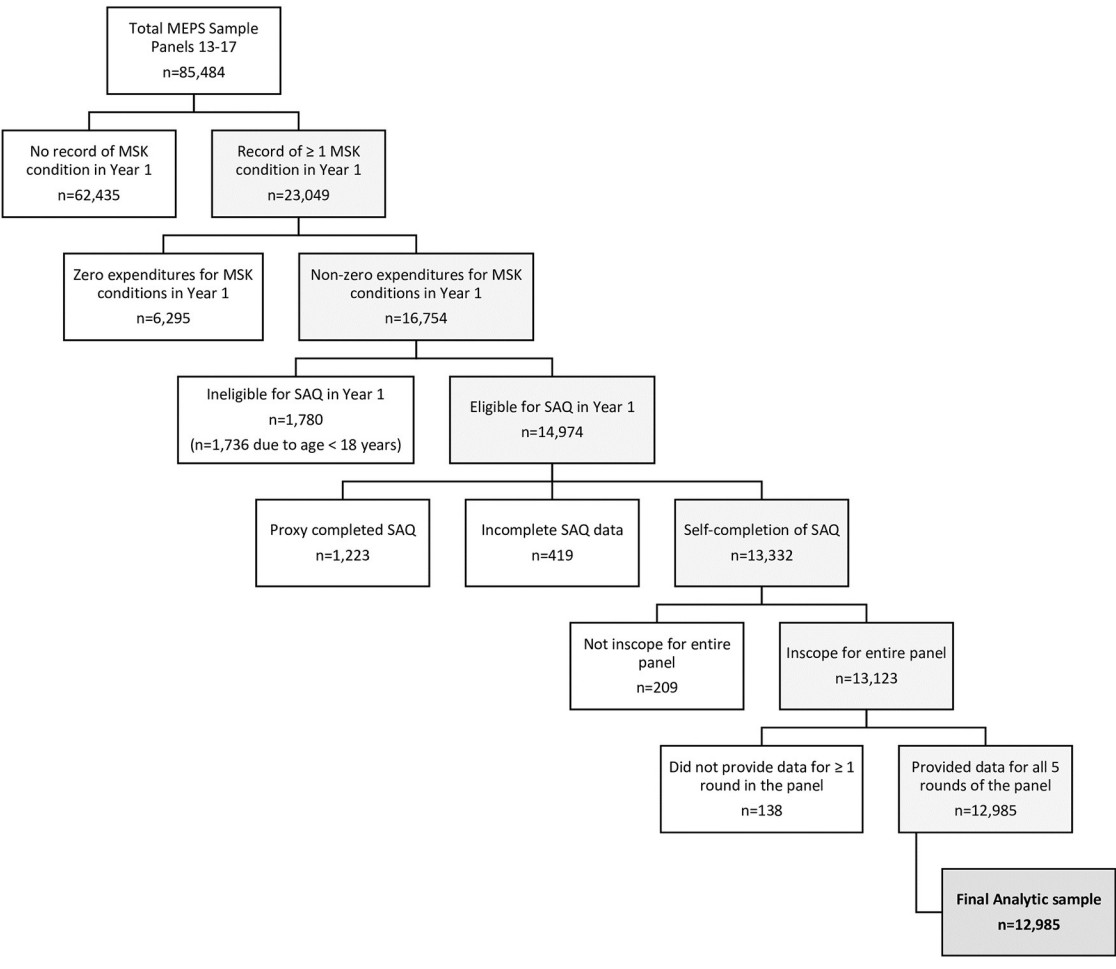

**Fig 1. Flow diagram of the selection process for the final analytic sample.**

musculoskeletal pain diagnoses (Table 3). After adjustment for non-modifiable variables and with the LOW group as the reference, modifiable variables associated with increased risk of being in the HIGH group were lower perceived physical and mental health status compared to others, higher levels of psychological distress and depression, greater days of work missed due to illness, higher pain interference, and greater number of prescription medications for pain. Modifiable variables associated with decreased risk of being in the HIGH group (i.e., protective against persistent high-cost utilization) were higher self-reported physical and mental health on the SF-12. Adjusted RRs with associated 95% confidence intervals are recorded in Table 3.

### Fully-adjusted multivariable analysis

All significant variables from the minimally-adjusted regression were included in the fully-adjusted model along with all non-modifiable variables. Examination of regression diagnostics demonstrated minimal concern for multicollinearity (all VIFs <10 and tolerances >0.1) among model variables. Assessment of fit demonstrated adequate performance of the multinomial regression model (Wald chi-square = 20.20, p < .001). When non-modifiable variables were included in the models the modifiable variables associated with increased risk of being in

**Table 1. Weighted percentile group means for demographic and health-related information.**

| Variable[a] | Low N = 1504[b] | Medium N = 10983[b] | High N = 498[b] | Total N = 12,985[b] | p-value |
|---|---|---|---|---|---|
| Age, yrs | 50.8 ± 0.7 (18–85) | 54.1 ± 0.3 (18–85) | 56.9 ± 0.8 (19–85) | 53.9 ± 0.3 (18–85) | < .001 |
| Body mass index | 28.7 ± 0.2 (14.6–70.8) | 28.9 ± 0.1 (14.1–77.3) | 29.5 ± 0.4 (16.0–72.0) | 28.9 ± 0.1 (14.1–77.3) | 0.15 |
| Charlson comorbidity index | 0.5 ± 0.01 (0–9) | 0.60 ± 0.01 (0–11) | 0.7 ± 0.1 (0–8) | 0.6 ± 0.01 (0–11) | 0.003 |
| PCS | 46.8 ± 0.4 (7.5–67.1) | 44.0 ± 0.2 (7.1–69.2) | 33.5 ± 0.6 (5.9–66.0) | 43.9 ± 0.2 (5.9–69.2) | < .001 |
| MCS | 50.01 ± 0.31 (7.1–78.1) | 50.0 ± 0.1 (5.2–75.0) | 45.4 ± 0.7 (13.1–70.5) | 49.9 ± 0.1 (5.2–78.1) | < .001 |
| General psychological distress | 4.1 ± 0.1 (0–24) | 4.3 ± 0.1 (0–24) | 7.0 ± 0.3 (0–24) | 4.4 ± 0.1 (0–24) | < .001 |
| Depression | 0.9 ± 0.04 (0–6) | 0.9 ± 0.02 (0–6) | 1.7 ± 0.1 (0–6) | 1.0 ± 0.02 (0–6) | < .001 |
| Total musculoskeletal conditions | 1.2 ± 0.02 (1–4) | 1.8 ± 0.02 (1–10) | 3.2 ± 0.1 (1–10) | 1.8 ± 0.02 (1–10) | < .001 |
| Total prescription drugs | 0.8 ± 0.03 (0–6) | 1.6 ± 0.03 (0–46) | 5.9 ± 0.3 (0–52) | 1.7 ± 0.04 (0–52) | < .001 |

[a] Data are mean ± standard deviation (range).

[b] Unweighted sample size.

PCS = physical component subscale of the SF-12, MCS = mental component subscale of the SF-12.

Low = annual pain-related expenditures in the lowest 15%; Medium = annual pain-related expenditures between 15–85%; High = annual pain-related expenditures in the highest 15%.

the HIGH compared to both the LOW and MEDIUM groups were higher number of missed work days and higher use of prescription medication for pain (Table 4). Greater pain interference was associated with higher risk of being in the HIGH compared to LOW group only. Modifiable variables associated with decreased risk of being in the HIGH group compared to both the LOW and MEDIUM groups were higher self-reported physical and mental health on the SF-12.

## Sensitivity analysis

Sensitivity analyses identified a range of population percentages for HIGH group membership between 2.13% (1.80–2.45%) and 6.03% (5.51–6.55%) when using the 90th and 80th percentile threshold definitions, respectively. Multivariable analyses using alternative thresholds for group membership demonstrated generally consistent findings with the primary analysis (S6 Table). Across all three analyses, age, insurance status, diagnosis type, and total number of musculoskeletal diagnoses were consistent non-modifiable variables associated with group classification. Among modifiable factors, sensitivity analyses supported the associations found in the primary analysis for self-reported physical health, pain interference, and use of prescription medication for pain. However, the relationship between number of missed work days and group classification differed from the primary analysis when the classification threshold was set to 10%. In this case, HIGH compared to LOW group classification was associated with higher risk of missed work, but not HIGH compared to MEDIUM. Neither sensitivity analysis supported the association between self-reported mental health on the SF-12 and group classification.

**Table 2. Demographic information for the sample (total) and each group (none, low, medium, high) with weighted population percentage estimates.**

| Variable[a] | | Low N = 1504[b] | Medium N = 10983[b] | High N = 498[b] | Total N = 12,985[b] | p-value |
|---|---|---|---|---|---|---|
| Sex | Male | 558 (39.0) | 4,026 (39.6) | 156 (35.4) | 4,740 (39.4) | 0.35 |
| | Female | 946 (61.0) | 6,957 (60.4) | 342 (64.6) | 8,245 (60.6) | |
| Race | White | 1,060 (82.2) | 8,164 (85.8) | 380 (87.7) | 9,604 (85.5) | < .001 |
| | Black | 340 (13.0) | 2,035 (9.3) | 87(8.4) | 2,462 (9.7) | |
| | Other | 104 (4.8) | 784 (4.9) | 31 (3.9) | 919 (4.8) | |
| Ethnicity | Hispanic | 310 (10.6) | 1,803 (8.2) | 79 (8.9) | 2,192 (8.5) | 0.004 |
| | Non-Hispanic | 1,194 (89.4) | 9,180 (91.8) | 419 (91.1) | 10,793 (91.5) | |
| Poverty category | Poor or Near Poor | 397 (16.8) | 2,508 (15.5) | 151 (20.9) | 3,056 (15.8) | < .001 |
| | Low Income | 266 (16.9) | 1,638 (12.9) | 73 (13.2) | 1,977 (13.3) | |
| | Middle Income | 433 (30.2) | 3,172 (28.3) | 126 (25.9) | 3,731 (28.4) | |
| | High Income | 408 (36.1) | 3,665 (43.3) | 148 (40.1) | 4,221 (42.4) | |
| Geographic region | Northeast | 210 (15.0) | 1,754 (18.4) | 83 (20.3) | 2,047 (18.1) | 0.07 |
| | Midwest | 353 (25.6) | 2,492 (23.7) | 101 (20.4) | 2,946 (23.8) | |
| | South | 563 (35.4) | 4,018 (34.8) | 170 (31.6) | 4,751 (34.7) | |
| | West | 378 (24.0) | 2,719 (23.2) | 144 (27.7) | 3,241 (23.5) | |
| Smoking status | Yes | 297 (19.5) | 2,001 (17.7) | 112 (22.3) | 2,410 (18.1) | 0.08 |
| | No | 1,170 (80.5) | 8,731 (82.2) | 376 (77.7) | 10,277 (81.9) | |
| Education | High school diploma or less | 1,017 (60.8) | 6,882 (57.2) | 317 (58.8) | 8,216 (57.7) | 0.11 |
| | Some college/college degree | 487 (39.2) | 4,101 (42.8) | 181 (41.2) | 4,769 (42.3) | |
| Employment | Employed or have a job to return to | 840 (58.0) | 5,805 (56.2) | 187 (41.9) | 6,832 (55.8) | < .001 |
| | Unemployed | 660 (42.0) | 5,147 (43.8) | 310 (58.1) | 6,117 (44.1) | |
| Pain interference | Not at all, a little bit, or moderately | 1,255 (86.0) | 7,965 (76.8) | 169 (43.1) | 9,389 (76.5) | < .001 |
| | Quite a bit or extremely | 230 (14.0) | 2,872 (23.2) | 320 (56.9) | 3,422 (23.5) | |
| Ability to overcome | Disagree strongly, disagree somewhat or uncertain | 1,209 (80.6) | 8,930 (82.3) | 434 (89.0) | 10,573 (82.4) | 0.01 |
| | Agree somewhat or agree strongly | 258 (19.4) | 1,743 (17.7) | 44 (11.0) | 2,045 (17.6) | |
| Insurance | Uninsured | 382 (20.4) | 1,879 (14.1) | 47 (8.8) | 2,308 (14.6) | < .001 |
| | Public Insurance | 424 (24.7) | 3,258 (25.4) | 214 (35.7) | 3,896 (25.7) | |
| | Private Insurance | 698 (54.8) | 5,846 (60.5) | 237 (55.5) | 6,781 (59.7) | |
| Physical health | Excellent, very good, or good | 1,134 (79.0) | 8,032 (78.0) | 242 (54.7) | 9,408 (77.2) | < .001 |
| | Fair or poor | 370 (21.0) | 2,942 (22.0) | 256 (45.3) | 3,568 (22.8) | |
| Mental health | Excellent, very good, or good | 1,368 (92.8) | 9,726 (90.4) | 383 (83.0) | 11,477 (90.4) | < .001 |
| | Fair or poor | 134 (7.2) | 1,253 (9.6) | 113 (17.0) | 1,500 (9.6) | |
| Usual care provider | Yes | 1,264 (85.7) | 9,694 (89.5) | 462 (94.6) | 11,420 (89.3) | < .001 |
| | No | 233 (14.3) | 1,232 (10.5) | 35 (5.4) | 1,500 (10.7) | |
| Missed ≥ 1 work day in year 1 | Yes | 156 (10.1) | 1,976 (17.5) | 115 (25.0) | 2,247 (17.0) | < .001 |
| | No | 1,348 (89.9) | 9,007 (82.5) | 383 (75.0) | 10,738 (83.0) | |
| Diagnosis | Musculoskeletal disease only | 1,195 (78.0) | 7,553 (67.4) | 305 (59.8) | 9,053 (68.2) | < .001 |
| | Musculoskeletal injury only | 226 (16.3) | 1,615 (15.0) | 27 (6.4) | 1,868 (14.8) | |
| | Musculoskeletal disease and Injury | 83 (5.6) | 1,815 (17.6) | 166 (33.8) | 166 (16.9) | |
| Metropolitan statistical area (MSA) | Non-MSA | 253 (18.7) | 1,803 (17.5) | 79 (15.5) | 2,135 (17.6) | 0.48 |
| | MSA | 1,251 (81.3) | 9,180 (82.4) | 419 (84.5) | 10,850 (82.4) | |

[a] Data are sample size (weighted population percentage estimates).

[b] Unweighted sample size.

Low = annual pain-related expenditures in the lowest 15%; Medium = annual pain-related expenditures between 15–85%; High = annual pain-related expenditures in the highest 15%.

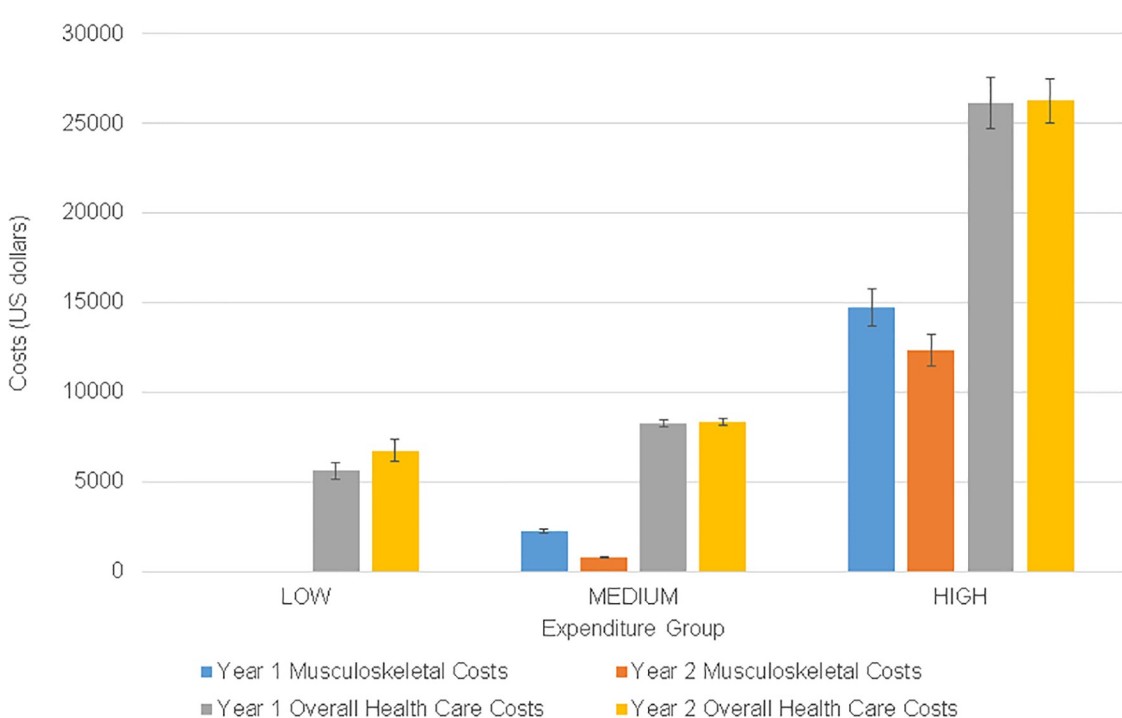

**Fig 2. Weighted mean annual expenditures for musculoskeletal pain diagnoses and all health care by expenditure group.** All costs are adjusted to 2013 U.S. dollars. Error bars are standard error of measure (SEM).

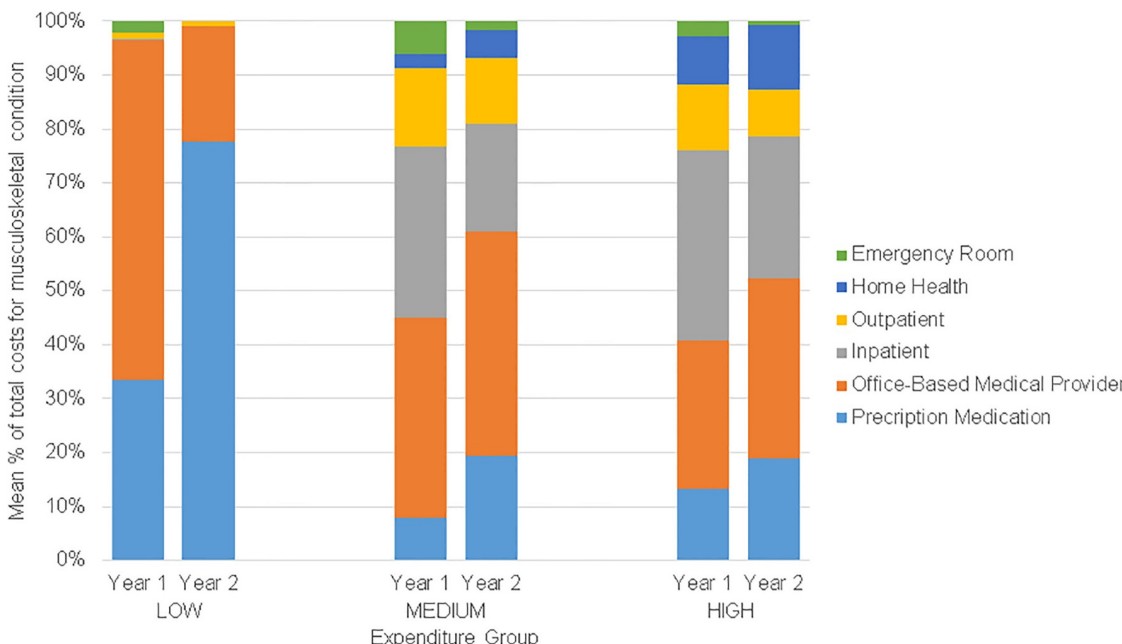

**Fig 3. Weighted mean percentage of total musculoskeletal pain costs attributable to each event type by expenditure group per year.**

**Table 3. Results of partially-adjusted regression models for modifiable variables (15% expenditure percentile criterion)[a].**

| Variable | Reference classification | Risk ratio[b,c] | 95% CI | |
|---|---|---|---|---|
| | | | Lower | Upper |
| BMI | Low | 0.99 | 0.98 | 1.02 |
| | Medium | 0.99 | 0.98 | 1.02 |
| Smoking status (Yes vs No) | Low | 1.28 | 0.88 | 1.83 |
| | Medium | 1.34 | 0.97 | 1.86 |
| Missed work days in Year 1 (missed ≥1-day vs no missed days) | Low | 3.75* | 2.33 | 5.99 |
| | Medium | 2.15* | 1.46 | 3.17 |
| PCS | Low | 0.94* | 0.92 | 0.95 |
| | Medium | 0.95* | 0.94 | 0.96 |
| MCS | Low | 0.97* | 0.96 | 0.98 |
| | Medium | 0.97* | 0.96 | 0.98 |
| Pain interference (Moderate, quite a bit or extreme vs a little bit or none) | Low | 5.10* | 3.76 | 6.94 |
| | Medium | 3.16* | 2.44 | 4.08 |
| General psychological distress | Low | 1.09* | 1.06 | 1.11 |
| | Medium | 1.08* | 1.05 | 1.10 |
| Depression | Low | 1.29* | 1.19 | 1.39 |
| | Medium | 1.23* | 1.15 | 1.31 |
| Perceived health status (Fair or poor vs good, very good or excellent) | Low | 2.11* | 1.55 | 2.89 |
| | Medium | 2.19* | 1.67 | 2.87 |
| Perceived mental health status (Fair or poor vs good, very good or excellent) | Low | 1.72* | 1.15 | 2.56 |
| | Medium | 1.35 | 0.96 | 1.90 |
| Ability to overcome (Disagree strongly, somewhat or uncertain vs agree somewhat or strongly) | Low | 1.52 | 0.90 | 2.57 |
| | Medium | 1.54 | 0.96 | 2.47 |
| Usual care provider (Yes vs No) | Low | 1.60 | 0.91 | 2.79 |
| | Medium | 1.51 | 0.88 | 2.59 |
| Total prescription medications in Year 1 | Low | 1.46* | 1.36 | 1.57 |
| | Medium | 1.22* | 1.16 | 1.28 |

[a] Parameters are for partially-adjusted models where each variable was considered individually after accounting for non-modifiable variables: age, sex, race, ethnicity, years of education, poverty category, employment status, metropolitan statistical area, census region, insurance status, number of comorbidities, diagnosis type and total number of musculoskeletal pain diagnoses.

[b] Risk ratios were adjusted for non-modifiable variables and reflect relative risk of being classified as HIGH compared to the reference classification. For continuous variables, values represent relative risk of being classified as HIGH compared to the reference classification for every one unit increase in the variable.

[c] Asterisk indicates $p<0.05$

Low = annual pain-related expenditures in the lowest 15%; Medium = annual pain-related expenditures between 15–85%; PCS = physical component subscale of the SF-12, MCS = mental component subscale of the SF-12.

## Discussion

Approximately 35% of the total pain-related costs in this study cohort were concentrated among the 4% of respondents with persistently high expenditures. By comparison, Rosella et al.[6] reported 55% of total health care costs concentrated within the top 5% of a sample of health care users in Canada, and Heslop et al.[8] reported 20% of total costs spent by 3% of hospital users in Australia. Even though the percentage of respondents is low, the absolute number meeting the criterion for persistently high pain-related expenditures in our study is quite large (i.e., almost 6 million individuals over the study timeframe) given the high prevalence of musculoskeletal pain.[3,4] Modifiable variables associated with persistent pain-related health care utilization may represent promising targets to reduce the societal and economic

**Table 4. Results of fully-adjusted multivariable logistic regression model (15% expenditure percentile criterion)[a].**

| Variable | Reference classification | Risk ratio[b,c] | 95% CI | |
|---|---|---|---|---|
| | | | Lower | Upper |
| *Non-modifiable variables* | | | | |
| Age | Low | 1.02* | 1.01 | 1.03 |
| | Medium | 1.01 | 0.99 | 1.02 |
| Sex (Female vs male) | Low | 0.81 | 0.59 | 1.11 |
| | Medium | 0.85 | 0.64 | 1.14 |
| Race (Black vs white) | Low | 0.58* | 0.41 | 0.84 |
| | Medium | 0.83 | 0.59 | 1.15 |
| Race (Other vs white) | Low | 0.72 | 0.35 | 1.47 |
| | Medium | 0.68 | 0.36 | 1.31 |
| Ethnicity (Hispanic vs non-Hispanic) | Low | 1.20 | 0.76 | 1.89 |
| | Medium | 1.31 | 0.85 | 2.04 |
| Education (Greater than high school vs high school or less) | Low | 1.46* | 1.01 | 2.09 |
| | Medium | 1.33 | 0.98 | 1.82 |
| Poverty level (Low income vs poor or near poor) | Low | 0.52* | 0.31 | 0.88 |
| | Medium | 0.72 | 0.44 | 1.16 |
| Poverty level (Middle income vs poor or near poor) | Low | 0.87 | 0.54 | 1.40 |
| | Medium | 0.89 | 0.57 | 1.37 |
| Poverty level (High income vs poor or near poor) | Low | 1.23 | 0.70 | 2.15 |
| | Medium | 1.11 | 0.69 | 1.78 |
| Employment (Unemployed vs employed) | Low | 0.98 | 0.62 | 1.55 |
| | Medium | 1.02 | 0.66 | 1.57 |
| Metropolitan Statistical Area (MSA) (Non-MSA vs MSA) | Low | 0.73 | 0.48 | 1.11 |
| | Medium | 0.81 | 0.57 | 1.15 |
| Census region (Midwest vs Northeast) | Low | 0.49* | 0.30 | 0.80 |
| | Medium | 0.65* | 0.44 | 0.97 |
| Census region (South vs Northeast) | Low | 0.62 | 0.37 | 1.03 |
| | Medium | 0.72 | 0.48 | 1.10 |
| Census region (West vs Northeast) | Low | 0.76 | 0.46 | 1.26 |
| | Medium | 0.96 | 0.64 | 1.43 |
| Charlson Comorbidity Index (CCI) | Low | 0.99 | 0.86 | 1.13 |
| | Medium | 0.97 | 0.87 | 1.08 |
| Insurance (Public vs uninsured) | Low | 2.43* | 1.48 | 3.97 |
| | Medium | 2.10* | 1.33 | 3.29 |
| Insurance (Private vs uninsured) | Low | 2.95* | 1.78 | 4.88 |
| | Medium | 2.05* | 1.31 | 3.23 |
| Diagnosis type (Injury vs disease only) | Low | 1.86* | 1.35 | 2.56 |
| | Medium | 1.15 | 0.89 | 1.50 |
| Total musculoskeletal conditions | Low | 3.14* | 2.62 | 3.79 |
| | Medium | 1.34* | 1.20 | 1.49 |
| *Modifiable variables* | | | | |
| Missed work days in Year 1 (missed ≥1-day vs no missed days) | Low | 2.49* | 1.53 | 4.05 |
| | Medium | 1.55* | 1.02 | 2.34 |
| PCS | Low | 0.96* | 0.94 | 0.98 |
| | Medium | 0.97* | 0.95 | 0.98 |
| MCS | Low | 0.98* | 0.97 | 0.99 |
| | Medium | 0.98* | 0.96 | 0.99 |

(*Continued*)

**Table 4.** (Continued)

| Variable | Reference classification | Risk ratio[b,c] | 95% CI | |
|---|---|---|---|---|
| | | | **Lower** | **Upper** |
| Pain interference (Moderate, quite a bit or extreme vs a little bit or none) | Low | 1.61* | 1.05 | 2.46 |
| | Medium | 1.19 | 0.82 | 1.73 |
| General psychological distress | Low | 0.98 | 0.92 | 1.04 |
| | Medium | 1.00 | 0.95 | 1.05 |
| Depression | Low | 1.04 | 0.89 | 1.23 |
| | Medium | 0.99 | 0.87 | 1.14 |
| Perceived health status (Fair or poor vs good, very good or excellent) | Low | 0.78 | 0.53 | 1.15 |
| | Medium | 1.03 | 0.72 | 1.46 |
| Perceived mental health status (Fair or poor vs good, very good or excellent) | Low | 1.13 | 0.68 | 1.87 |
| | Medium | 0.84 | 0.55 | 1.29 |
| Total prescription medications in Year 1 | Low | 1.37* | 1.27 | 1.48 |
| | Medium | 1.18* | 1.13 | 1.24 |

[a] Model fit statistics: Generalized Logit model Wald Chi-square test = 20.20, p < .001.

[b] Risk ratios were adjusted for all modifiable and non-modifiable variables included in the model and reflect relative risk of being classified as HIGH compared to the reference classification. For continuous variables, values represent relative risk of being classified as HIGH compared to the reference classification for every one unit increase in the variable.

[c] Asterisk indicates p<0.05

Low = annual pain-related expenditures in the lowest 15%; Medium = annual pain-related expenditures between 15–85%; PCS = physical component subscale of the SF-12, MCS = mental component subscale of the SF-12.

impact of many common musculoskeletal pain conditions. The specific modifiable variables identified in this study include self-reported physical and mental health, pain interference, and use of prescription medication for pain. Persistently high costs were also associated with higher number of missed work days, which previous studies have indicated can be beneficially modified through early workplace re-engagement programs.[38,39] Sensitivity analyses supported the associations found in the primary analysis for self-reported physical health, pain interference, and use of prescription medication for pain. Delivery models that prospectively identify and address these modifiable factors may have the greatest potential to impact costs and should be evaluated in future studies for their ability to influence value of care for musculoskeletal pain.[2]

This study did not evaluate whether services used were guideline-adherent or met the respondent's pain management needs. Therefore we cannot assert that those with persistently high costs should be using fewer services, had avoidable expenditures, or received care that did not adhere to evidence-based clinical guidelines. Nevertheless, treatment associated with persistence of high costs is often scrutinized in value-based health care systems and payment models. Identifying characteristics related to persistence of high costs provides initial direction for subsequent health care research and policy that aims to reduce the risk of high costs, particularly when higher costs are not associated with proportionally better outcomes.

Health care delivery models that address physical and mental health-related factors like physical functioning, general health perceptions, energy and vitality, social functioning, and role limitations due to physical and emotional problems may help to reduce the risk of persistently high pain-related utilization and costs. For those who are limited in their work capacity due to pain, treatments may also need to emphasize return to work training (e.g., training in work-related tasks, general physical conditioning, or cognitive behavioral theory-based strategies to improve pain coping), which is consistent with prior studies from occupational settings

demonstrating the benefits of interventions aimed at timely return to work.[48] Our results additionally suggest the need to better understand the downstream effects of high prescription medication use given its association with higher subsequent pain-related expenditures even after accounting for pain interference and psychological characteristics. While our study design does not allow for causal inference, it does indicate that high utilization of prescription medication for pain (regardless of the type of medication) is associated with higher spending and deserves further study to determine when pharmacological management delivers the highest value, or when non-pharmacological treatments might be more cost-effective over the entire course of treatment. Finally, while non-modifiable characteristics were not the primary focus of this analysis because they cannot be directly targeted through treatment, they are often important moderators of treatment outcomes.[49,50] Therefore, clinicians should consider the potential benefits of tailoring interventions based on relevant characteristics such as level of education, insurance coverage, or concomitant musculoskeletal conditions. Future research is needed to determine whether treatment approaches addressing both modifiable (e.g. prescription medication) and non-modifiable factors (e.g., socioeconomic status) result in expenditures aligned with appropriate care, better outcomes, and higher value for those at highest risk.

This work provides novel information that advances the study of high-cost utilization in a high-priority population that commonly seeks health care services.[7,8,41,51] Our findings converge with prior studies in other health conditions and health care systems which have consistently shown that older age, multiple chronic conditions, poorer self-perceived health, and lower socioeconomic status are associated with high-cost utilization.[6,7,45,52–55] Our results also compare favorably to studies in the U.S. that examine high-cost utilization at a single point in time, with higher expenditures among those who are insured.[56] Insurance coverage affords greater access to care in the U.S., which can contribute to higher expenditures, but does not necessarily protect against receiving low-value services.[57] An important direction for future research is to identify high-cost services that do not meaningfully improve pain-related outcomes, but are delivered more commonly because the service is highly available and/or an individual is insured for the service.

Psychological distress and depression were associated with persistent high costs for musculoskeletal pain in partially-adjusted multivariable analyses, but their influence was diminished in the fully-adjusted multivariable analyses. This suggests that other measures might adequately account for distress and depression (e.g., the SF-12 mental component sub-score) when modeling cost-related outcomes. Another reason for this finding is that other variables fully mediate the relationship between these psychological characteristics and health care utilization. Importantly, our models controlled for variables commonly used as clinical outcomes in studies on pain (e.g., pain interference, self-reported function). Therefore, psychological characteristics might be useful when considered in isolation or when evaluating factors related to clinical outcomes. However, their association with high utilization and expenditures for musculoskeletal pain is attenuated when considered along with other pain-related variables (e.g., pain interference).

Our sample had a higher proportion of unemployed respondents (44.1%) than published US population data from this time (34%; National Health Interview Study (NHIS), 2012).[58] In this study, unemployed respondents included those who were retired and/or out of work for any reason during the entire reference period. This higher proportion of unemployment may be due, in part, to the older average age of our sample (and therefore larger proportion of retirees) compared the general US population (median 54.1 years in our sample versus 37.6 years in the US population in 2013)[59], as well as the number of adults who are out of the workforce or disabled due to musculoskeletal pain conditions. Across the entire sample,

approximately 17% reported one or more missed work day. When considering only those individuals who report employment, the incidence of one or more missed work days was approximately 31%. This rate is slightly lower compared to NHIS employment data, which shows 38–40% of the US workforce experiencing one or more missed work days.[58,60] This finding seems counterintuitive since we might expect those with pain conditions to have a higher incidence of missed work days compared to the general population. One explanation for the discrepancy could be that the incidence of one or more missed work day is lower among older adults in the US compared to the general population, even though older working adults tend to have a higher average number of missed work days.[61] Thus the higher proportion of older adults in our sample could help explain the lower incidence of a missed work day.[61]

This study focused on persistence of high costs for musculoskeletal pain as a first step towards establishing value based care parameters. However we acknowledge that unmet health care needs attributable to limited access to care, insurance constraints or financial barriers among those with low or no costs represent equally important endpoints. Given our study design, we were not able to determine whether those who did not receive care, or had low costs, simply required less care to effectively manage their condition or had unmet health care needs. This issue will be an important one to address in future studies. Studies that evaluate care-seeking decisions, appropriateness of care received for the cost, or absolute barriers to care address fundamentally different research questions than what we proposed in this study. But these questions are nonetheless important to improve the value of care for musculoskeletal pain. While identifying predictors of LOW expenditure group membership was not the primary focus of this analysis, characteristics that differentiate this group, like race and insurance status, may indicate where unmet needs and limited access to care are likely a concern for those with musculoskeletal conditions. These findings provide motivation for future studies that aim to reduce health care disparities among those with musculoskeletal pain conditions and improve access to evidence-based treatments.

Strengths of this study include a robust dataset with variables representing a variety of predisposing and enabling factors, as well as other perceived and evaluated need characteristics. We focused on identifying modifiable factors but also evaluated non-modifiable factors since both have relevance for designing health care interventions and policy. Some factors such as missed work days and prescription pain medication use could be alternatively classified as non-modifiable since they are often viewed as the result of a condition, and not something that can be easily manipulated. However, prior research in occupational settings[38,39] suggests these can be important treatment targets and there remains the potential that intervening on these variables could impact downstream utilization and costs in other patient populations. An additional strength is that we used a robust, population-based data set with survey weights and statistical procedures that account for differential sample selection probabilities and adjust for nonresponse and loss to follow-up. In essence, this allowed us to derive estimates consistent with what we would expect to find had we been able to survey the entire U.S. noninstitutionalized civilian population. The benefit of this approach is that we can use our findings to more confidently inform broader population-level health care research and policy. We also took multiple steps to assess the sensitivity of our findings, by alternately defining high-cost utilization for musculoskeletal pain. As a result, we have confidence that significant variables consistently identified across analyses are most likely to be reproduced in future studies.

This analysis also had some limitations. First, while this study was able to capture costs and utilization over the course of 2 years, our ability to model longitudinal spending was limited. Moreover, our study design prevented us from inferring causal relationships between model variables and persistence of health care costs. However, our findings can inform variable selection in future persistent high cost utilization studies that are specifically designed to establish

causality and examine longer term outcomes. Second, we excluded those who had proxy responses for the SAQ since proxy responses may not reflect true subjective experiences and could produce unreliable results.[21–23] Individuals with proxy responses tended to be older and in poorer physical and mental health. A small percentage of respondents had incomplete follow up data, largely due to death or becoming institutionalized. These individuals were also older and had poorer mental and physical health. Exclusion of these respondents may limit generalizability, which should be considered when interpreting our results. Specifically, our findings may not be generalizable to older adults with extremely poor physical or mental health, those who are near death, and/or those who might require a proxy to report their level of pain, function, disability, or health care use.

A third limitation is that the survey design did not allow for assessment of self-reported measures to coincide temporally with onset of the injury or condition. All demographic and health-related variables were collected from the earliest data collection point in the panel, and in some cases may have been measured prior to the episode of care. This could limit the utility of these variables in future clinical prediction models, and their prediction potential should be confirmed in prospective studies. Fourth, some expenditures are not included in the MEPS survey, such as over-the-counter medications and durable medical equipment. Indirect costs, such as lost wages, are also not included. The direct method of expenditure calculation used in these analyses is likely to substantially underestimate total costs for musculoskeletal pain.[62] Therefore, these results are most pertinent to stakeholders interested in reducing direct pain-related costs among those seeking health care for musculoskeletal pain. Our findings may also not be generalizable to some specific patient groups, such as children or adolescents. Chronic pain as an adult can begin in childhood for many individuals. However, assessment methods, approaches to treatment, and health care utilization patterns for pediatric and adolescent populations may differ compared to adults, making results of an analysis that combines adult and pediatric populations potentially difficult to interpret and apply. More practically, SAQ data were not available for anyone under the age of 18, which would require removing many important self-reported variables from the analysis.

Fifth, we excluded those without costs in Year 1 since we were interested in identifying factors associated with persistent costs. This means that these results are specific to those that were seeking care for their musculoskeletal pain condition. These findings cannot be applied to those that did not seek care. Finally, we used data from 2008–2013, which pre-dated the switch from ICD-9 to ICD-10 diagnostic codes in the US. We chose this approach to limit the potential influence that variation in coding could have on determining year-to-year spending. Limited MEPS data are currently available to track longitudinal outcomes following the change to ICD-10. Interim changes in health care policy and reimbursement since 2013 could impact the application of these findings in the current health care system. However, many of the variables identified in this study continue to be relevant drivers of health care use and spending in other populations because our findings converge with studies using more recent data.[45,55]

## Conclusion

Among patients with musculoskeletal pain, greater than one-third of direct health care costs are concentrated among a small percentage (i.e. less than 5%) of individuals defined as persistent high-cost utilizers. Demographic, health, and pain-related characteristics can help identify these individuals. Health care delivery models that prospectively identify individuals at-risk of being high-cost utilizers for musculoskeletal pain conditions and address modifiable risk factors may improve costs, and potentially health care value. Future studies are needed to better

understand the circumstances in which persistent high costs are appropriate outcomes for management of musculoskeletal pain versus when they may be indicative of low value care.

## Supporting information

**S1 Appendix. Definition and analysis coding of categorical variables.**
(DOCX)

**S1 Table. List of musculoskeletal ICD-9 code diagnoses.**
(DOCX)

**S2 Table. Weighted means for demographic and health-related information that differed significantly between groups.** [a] Data are mean ± standard error of mean (range). [b] Unweighted sample size. PCS = physical component subscale of the SF-12, MCS = mental component subscale of the SF-12.
(DOCX)

**S3 Table. Sample distributions with weighted population percentage estimates for demographic and health-related information that differed significantly between groups.** [a] Data are sample size (weighted population percentage estimates). [b] Unweighted sample size.
(DOCX)

**S4 Table. Unweighted frequency table of diseases of the musculoskeletal system and connective tissue ICD-9 codes.**
(DOCX)

**S5 Table. Unweighted frequency table of musculoskeletal injury ICD-9 codes.**
(DOCX)

**S6 Table. Results of fully-adjusted multivariable logistic regression model sensitivity analyses (10% and 20% expenditure percentile criterion).** [a] Model fit statistics: Generalized Logit model Wald Chi-square test = 20.20, p < .001. [b] Adjusted odds ratios reflect odds of being classified as HIGH compared to the reference classification. For continuous variables, values represent odds of being classified as HIGH compared to the reference classification for every one unit increase in the variable. Low = annual pain-related expenditures in the lowest 15%; Medium = annual pain-related expenditures between 15–85%; PCS = physical component subscale of the SF-12, MCS = mental component subscale of the SF-12.
(DOCX)

## Author Contributions

**Conceptualization:** Trevor A. Lentz, Jeffrey S. Harman, Nicole M. Marlow, Jason M. Beneciuk, Roger B. Fillingim, Steven Z. George.

**Data curation:** Trevor A. Lentz.

**Formal analysis:** Trevor A. Lentz, Jeffrey S. Harman, Nicole M. Marlow, Jason M. Beneciuk.

**Funding acquisition:** Trevor A. Lentz, Steven Z. George.

**Investigation:** Trevor A. Lentz, Jeffrey S. Harman, Nicole M. Marlow, Jason M. Beneciuk, Roger B. Fillingim, Steven Z. George.

**Methodology:** Trevor A. Lentz, Jeffrey S. Harman, Nicole M. Marlow, Jason M. Beneciuk, Roger B. Fillingim, Steven Z. George.

**Project administration:** Trevor A. Lentz.

**Resources:** Steven Z. George.

**Software:** Steven Z. George.

**Supervision:** Jeffrey S. Harman, Nicole M. Marlow, Jason M. Beneciuk, Roger B. Fillingim, Steven Z. George.

**Validation:** Trevor A. Lentz, Nicole M. Marlow, Jason M. Beneciuk, Roger B. Fillingim, Steven Z. George.

**Visualization:** Trevor A. Lentz.

**Writing – original draft:** Trevor A. Lentz, Jeffrey S. Harman, Nicole M. Marlow, Jason M. Beneciuk, Roger B. Fillingim, Steven Z. George.

**Writing – review & editing:** Trevor A. Lentz, Jeffrey S. Harman, Nicole M. Marlow, Jason M. Beneciuk, Roger B. Fillingim, Steven Z. George.

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
