## [Decision Letter · Decision Letter 0]

23 Jul 2019

PONE-D-19-18187

Factors Associated with Persistently High-Cost Health Care Utilization for Musculoskeletal Pain

PLOS ONE

Dear Dr. Lentz,

Thank you for submitting your manuscript to PLOS ONE. After careful consideration, we feel that it has merit but does not fully meet PLOS ONE’s publication criteria as it currently stands. Therefore, we invite you to submit a revised version of the manuscript that addresses the points raised during the review process.

Below you will find the comments from two independent reviewers, as well as my own comments on your submission. Please provide further justification for the research methods, and consider revising parts of the analytic approach and interpretation.

We would appreciate receiving your revised manuscript by Sep 06 2019 11:59PM. To enhance the reproducibility of your results, we recommend that if applicable you deposit your laboratory protocols in protocols.io, where a protocol can be assigned its own identifier (DOI) such that it can be cited independently in the future. For instructions see: http://journals.plos.org/plosone/s/submission-guidelines#loc-laboratory-protocols

We look forward to receiving your revised manuscript.

Kind regards,

Melita J. Giummarra

Academic Editor

PLOS ONE

Journal Requirements:

Additional Editor Comments:

Overall this is an interesting and valuable study. I noted several aspects of the methods and analyses that require attention to ensure clear interpretation from readers, and potential future replication of the study.

Please justify excluding cases with a  proxy response for the adult SAQ (e.g., is it not validated for a proxy response?), and why you chose not to use imputation methods to include cases with incomplete SAQ responses.The STROBE chart should be included in the body of the paper. On this figure it is not clear what data are being shown in the parentheses.Can you please explain the “condition-event crosswalk” in more detail.How is “poverty category” calculated or defined?To improve clarity for an international audience can a little more detail be provided on the different types of health insurance?The analyses used multinomial logistic regression, but did you check whether the outcome was suitable for ordinal regression? Assuming that it was not ordinal, can you also indicate whether the assumption of the independence of irrelevant alternatives was met. Also, the results report odds ratios, but a multinomial logistic regression yields a risk ratio. Please check and clarify.The results should include an analysis of differences between those excluded from analysis due to loss to follow up. This is particularly relevant given that a loose statement on potential biases due to participant exclusion in the discussion (lines 444-446). If these characteristics were analysed you could comment on what those biases are.The results should only be reported to two decimal places in the tables.The results identify a range of modifiable variables associated with pain-related healthcare expenditure, but can you be sure about the direction of the relationship with MSK conditions/cost of treatment? Please comment on this in the discussion.People who were insured had higher expenditure, but this suggests that having insurance facilitates healthcare access and funding. Can you comment on potential unmet treatment needs in those with low costs?I agree with the reviewers that excluding those with no healthcare costs at follow up seems like an odd omission if you are seeking to understand the scope of the problem at a population level. Couldn't you combine those with none and low healthcare expenditure, or have an additional group with $0 costs?

Reviewers' comments:

Reviewer's Responses to Questions

**Comments to the Author**

1. Is the manuscript technically sound, and do the data support the conclusions?

Reviewer #1: Yes

Reviewer #2: Partly

2. Has the statistical analysis been performed appropriately and rigorously? 

Reviewer #1: Yes

Reviewer #2: Yes

3. Have the authors made all data underlying the findings in their manuscript fully available?

Reviewer #1: Yes

Reviewer #2: Yes

4. Is the manuscript presented in an intelligible fashion and written in standard English?

Reviewer #1: Yes

Reviewer #2: Yes

5. Review Comments to the Author

Reviewer #1: General comments:

This is a very well-written manuscript representing a very important health care topic, high costs associated with management of musculoskeletal conditions. The use of the Medical Expenditures Panel Survey is an appropriate and efficient data source. The findings of the analysis are important for understanding of this problem. Of great importance is the finding that 4% of the population are responsible for 35% of total costs. Minor clarifications as specified below are requested.

Specific comments:

Need further elaboration of how days of work missed due to illness is a modifiable factor. This seems more like an outcome.

What is meant regarding the population of the study representing a very large population (Page 15), 12.985 represents 150,792,698 individuals?

The mean and median costs for the low and medium expenditure groups seems surprisingly low. $23 for the low group wouldn’t even cover an office visits. For the medium groups, $1551 may be reasonable but even that seems low. Please comment on this – is it possible that pain-related care is coded as something else?

The BMI means seem low and not representative of the overweight and obesity levels of the US population. It would be interesting to see the distribution (%) within the CDC weight classifications.

Do number of prescriptions in table 1 include pain medications and non-pain medications? Does this refer to individual prescriptions filled or number of unique drugs per subject?

There is a large proportion of subjects that are not employed, 45%. Is days missed from work relevant? Is this internally valid? That is, for persons who are not employed are there reported days missed from work?

It is surprising that such a large proportion of the overall (77%) and 55% of the high utilizers report excellent, very good and good physical health? It might be useful to examine this as more than just a dichotomy with excellent, very good and good vs. fair or poor.

Related to previous comment, with 25% of persons in the high utilizer group reporting missing one or more days of work and 17% overall seems low – is this skewed by the large proportion of persons not employed? Can this be compared to other national data on missed time from work?

Reviewer #2: This paper confirms the findings of previous studies in other countries with single payer systems that the highest resource users with musculoskeletal conditions represent a small proportion of people with the condition.

The paper conducts a statistical analysis of the factors associated with high resource use. The paper is well written and clear and put in the context of prior studies.

The paper is somewhat let down by the implied interpretation of the findings. That is, that while many patients do not receive the most effective and evidence based care, that this is a particular issue for those who are the highest resource users and that using more "value-based" approaches would reduce the high proportion of expenditure on the highest usage group. However it cannot be concluded from this study that the highest resource users are the most likely to be receiving more inappropriate care than any other patients, nor that use of evidence based approaches would reduce the cost for this group.

A more accurate assessment from this study might be:

- Being in the poorest health and poor (as well as pain interfering most with the patient's life) is associated with being in the highest cost group, as noted by the authors

- When pain persists so does the expenditure

- Many patients in other studies have been shown to receive less that optimal care without a good evidence base

- Access to more evidence based approaches may improve health outcomes for some patients

- Use of the most effective care may or may not reduce cost - this is not clear from this study.

- Pain is not a one-size-fits-all condition, thus the most effective treatment needs to be found for each patient. Patient -centred care is important as pain is often complex and multifaceted and difficult to treat effectively.

General comments include:

- Was the number of musculoskeletal conditions and number and type of co-morbidities considered in the analysis as predictors of high resource use?

- Why exclude patients <18 years of age, as chronic pain can start in childhood?

- Why exclude patients who reported a musculoskeletal condition but no related expenditure, they equally may have not been receiving the most appropriate care? Were these patients a lower income group with less access to care?

- Comorbidities - can more detail be given on what is included in the index as this is an important predictor of resource use?

- The self efficacy assumption seems to be that the highest cost group should be able to manage their own condition. However there is no evidence for this and it seems very unlikely that the sickest group could manage their own pain without qualified medical support, given finding the optimal treatment is challenging even for specialist pain physicians.

- p 11 lines 198 and 199 seems to imply that the highest cost users are likely abusing prescription pain medication, or at least this is more likely in the highest resource use group, but there is no evidence for this.

- p 13 Why allocate patients with zero expenditure in year 2 as low or middle - why not use total expenditure over 2 years?

- How much of expenditure is predicted by severity of pain alone?

6. PLOS authors have the option to publish the peer review history of their article (what does this mean?). If published, this will include your full peer review and any attached files.

Reviewer #1: No

Reviewer #2: Yes: Professor Deborah Schofield

---

## [Author Response · Author response to Decision Letter 0]

21 Aug 2019

Please see attached response to reviewers document.

---

## [Decision Letter · Decision Letter 1]

16 Sep 2019

PONE-D-19-18187R1

Factors Associated with Persistently High-Cost Health Care Utilization for Musculoskeletal Pain

PLOS ONE

Dear Dr. Lentz,

Thank you for submitting your manuscript to PLOS ONE. After careful consideration, we feel that it has merit but does not fully meet PLOS ONE’s publication criteria as it currently stands. Therefore, we invite you to submit a revised version of the manuscript that addresses the points raised during the review process.

Thank you for addressing nearly all comments raised from the first review of your submission. There were a couple of points that still need to be addressed. First, thank you for the explanation for why an ordinal model was not used, but can you please provide further information in the manuscript on whether the assumptions for a multinomial model were met (i.e., the IIA). Second, you have highlighted that you chose to compare the characteristics/biases of the respondents excluded due to proxy response vs those that were included, but I did not see these results reported in the results section of the paper.

We would appreciate receiving your revised manuscript by Oct 31 2019 11:59PM. To enhance the reproducibility of your results, we recommend that if applicable you deposit your laboratory protocols in protocols.io, where a protocol can be assigned its own identifier (DOI) such that it can be cited independently in the future. For instructions see: http://journals.plos.org/plosone/s/submission-guidelines#loc-laboratory-protocols

A rebuttal letter that responds to each point raised by the academic editor and reviewer(s). This letter should be uploaded as separate file and labeled 'Response to Reviewers'. In your response letter, please indicate the specific page and/or line of your revision where the changes have been executed. Make sure that the line numbers are carried throughout the whole manuscript if choosing to refer to line numbers.A marked-up copy of your manuscript that highlights changes made to the original version. This file should be uploaded as separate file and labeled 'Revised Manuscript with Track Changes'.An unmarked version of your revised paper without tracked changes. This file should be uploaded as separate file and labeled 'Manuscript'.

We look forward to receiving your revised manuscript.

Kind regards,

Melita J. Giummarra

Academic Editor

PLOS ONE

Additional Editor Comments (if provided):

Please move discussion of the results to the discussion (e.g., the proportion of unemployed respondents compared with the US population).

Reviewers' comments:

Reviewer's Responses to Questions

**Comments to the Author**

1. If the authors have adequately addressed your comments raised in a previous round of review and you feel that this manuscript is now acceptable for publication, you may indicate that here to bypass the “Comments to the Author” section, enter your conflict of interest statement in the “Confidential to Editor” section, and submit your "Accept" recommendation.

Reviewer #1: All comments have been addressed

Reviewer #2: (No Response)

2. Is the manuscript technically sound, and do the data support the conclusions?

Reviewer #1: Yes

Reviewer #2: Partly

3. Has the statistical analysis been performed appropriately and rigorously? 

Reviewer #1: Yes

Reviewer #2: I Don't Know

4. Have the authors made all data underlying the findings in their manuscript fully available?

Reviewer #1: Yes

Reviewer #2: Yes

5. Is the manuscript presented in an intelligible fashion and written in standard English?

Reviewer #1: Yes

Reviewer #2: Yes

6. Review Comments to the Author

Reviewer #1:

This is a very well-written manuscript representing a very important health care topic, high costs associated with management of musculoskeletal conditions.  The use of the Medical Expenditures Panel Survey is an appropriate and efficient data source.  The findings of the analysis are important for understanding of this problem.  Of great importance is the finding that 4% of the population are responsible for 35% of total costs.   A minor explanation/clarification as specified below is requested.

Though I do say that all previous comments have been requested, the issue of extrapolation of the study population to the larger population it is intended to represent would be helpful.

Specific comment/question:

Line 307 – How the 12,985 study subjects was identified is clearly described.  In line 307 it is stated that these subjects represent over 150 million unique individuals – please explain.  I asked this question on the initial review, also.  This same issue occurs in lines 333 and 335 when the high and low cost subsets are described.

Reviewer #2: 

This is a well written paper on the higher service use group with musculoskeletal pain. I have a number of comments:

Overall: It is not surprising if people in the most pain and with the greatest pain interference use more health services. My main concern with the paper is the implicit suggestion that the high service use group should be using less services. However the paper does not contain the required information to determine whether the services used comply with current guidelines or best meet an individual patient’s pain management needs. It is important to remember that for some types of pain e.g. neuropathic pain, patients report a quality of life worse than death. See for example: Torrance et al (2014) Pain 155(10):1996-2004 doi: 10.1016/j.pain.2014.07.001. The biggest challenge in the field of chronic pain is the lack of effective treatment for many patients and the lack of access to the most effective treatments for others. This needs to be drawn out in the paper.

Specific comments:

Suggest replacing the use of the word persistent with a more neutral term such as continuous. Persistent has a connotation that the patients are a nuisance.

Methods:

Page 6 line 72: What are the conditions where expenditures are not expected to be persistently high and how was this definition arrived at? The severity and duration of pain is not uniform or easily predictable.

P 7 lines 96-98 Are these expenditures specifically to treat chronic pain?

P 10 lines 157 – 162 Asking patients if they feel they can overcome their illness without help from a medically trained person is rather odd in this context. Why should patients be able to do this when many clinicians find managing pain effectively difficult. Would the authors suggest that cancer or CVD patients should be able to manage their health condition without help from a medically trained person?

P 12 lines 207-8 Use of opiods: There is now quite a large body of literature on overcoming medication nonadherance (also known as noncompliance)? Why is adherence to medical advice seen as a poor outcome in this paper?

Page 14 The high use group is the top 15% but the abstract refers to them as the top 4%. Can the authors please clarify?

Results:

P 17 line 315 Does unemployed also include retired and not in the labour force for other reasons?

P 17 lines 319-330 This is confusing. Is the conclusion that patients in chronic pain take fewer days off work, and those who are older and in pain take even fewer days off work? If so, this is a remarkable and important conclusion that should be drawn out in the discussion. Were patients not in the labour force excluded from this analysis? Were the results adjusted for age?

P 22 lines 358-368 Highest use of services is amongst those who report the poorest physical and mental health – isn’t that to be expected? Was pain severity considered? How exactly is poorer health to be modified?

Discussion:

P 27 While the treatments suggested may be beneficial, the median age of the sample for this study is 54 years with 44% unemployed. What treatments would benefit the group not in employment?

P 28 “Downstream effects of high prescription use”: The text implies that high use of medication causes the downstream effects - this needs to be more clearly described as an association not effects.

Perhaps the greatest need in relation to improved care of musculoskeletal pain conditions is for effective treatments for chronic pain to be accessible and affordable.

7. PLOS authors have the option to publish the peer review history of their article (what does this mean?). If published, this will include your full peer review and any attached files.

Reviewer #1: No

Reviewer #2: No

---

## [Author Response · Author response to Decision Letter 1]

28 Oct 2019

See attached response to reviewers

---

## [Editor Report · Decision Letter 2]

30 Oct 2019

Factors Associated with Persistently High-Cost Health Care Utilization for Musculoskeletal Pain

PONE-D-19-18187R2

Dear Dr. Lentz,

We are pleased to inform you that your manuscript has been judged scientifically suitable for publication and will be formally accepted for publication once it complies with all outstanding technical requirements.

With kind regards,

Melita J. Giummarra

Academic Editor

PLOS ONE
---

## [Editor Report · Acceptance letter]

1 Nov 2019

PONE-D-19-18187R2 

Factors Associated with Persistently High-Cost Health Care Utilization for Musculoskeletal Pain 

Dear Dr. Lentz:

I am pleased to inform you that your manuscript has been deemed suitable for publication in PLOS ONE. Congratulations! Your manuscript is now with our production department. 

With kind regards,

on behalf of

Dr. Melita J. Giummarra 

Academic Editor

PLOS ONE